# Invariance and identifiability issues for word embeddings

**Rachel Carrington**     **Karthik Bharath**     **Simon Preston**

School of Mathematical Sciences, University of Nottingham

`{rachel.carrington, karthik.bharath, simon.preston}@nottingham.ac.uk`

## Abstract

Word embeddings are commonly obtained as optimizers of a criterion function $f$ of a text corpus, but assessed on word-task performance using a different evaluation function $g$ of the test data. We contend that a possible source of disparity in performance on tasks is the incompatibility between classes of transformations that leave $f$ and $g$ invariant. In particular, word embeddings defined by $f$ are not unique; they are defined only up to a class of transformations to which $f$ is invariant, and this class is larger than the class to which $g$ is invariant. One implication of this is that the apparent superiority of one word embedding over another, as measured by word task performance, may largely be a consequence of the arbitrary elements selected from the respective solution sets. We provide a formal treatment of the above identifiability issue, present some numerical examples, and discuss possible resolutions.

## 1  Introduction

Word embeddings map a text corpus, say $X$, to a collection of vectors $V = (v_1, ..., v_p)$ where each $v_j \in \mathbb{R}^d$, for a prescribed embedding dimension $d$, represents one of $p$ words in the corpus. Different word embedding models can be cast as the solution of an optimisation

$$\underset{U,V}{\arg\min}\ F(X, U, V) = \underset{U,V}{\arg\min}\ f(X, UV), \tag{1}$$

for particular corpus representation $X$ and objective function $f$, where $U = (u_1, \ldots, u_n)^T$ are vectors in $\mathbb{R}^n$ representing contexts, typically not of main interest. The setup subsumes some popular embedding techniques such as Latent Semantic Analysis (LSA) [Deerwester et al., 1990], word2vec [Mikolov et al., 2013b,a], and GloVe [Pennington et al., 2014], wherein the matrices $U$ and $V$ appear in a suitably chosen $f$ only through their product $UV$.

Once a word embedding $V$ is constructed by solving (1), the embedding is evaluated on its performance in tasks, including identifying word *similarity* (given word $a$, identify words with similar meanings), and word *analogy* (for the statement "$a$ is to $b$ what $c$ is to $x$", given $a$, $b$ and $c$, identify $x$). Similarities or analogies can be computed from $V$, then performance evaluated against a test data set $D$ containing human-assigned judgements as

$$g(D, V), \tag{2}$$

for some function $g$. Constructing word embeddings is "unsupervised" with respect to the evaluation task in the sense that $V$ is determined from (1) independently of the choice of $g$ and the data $D$ in (2), although $f$ typically entails free parameters that may, consciously or not, be chosen to optimize (2) [Levy et al., 2015].

Different word embedding models, identified as different $f$ in (1), are often compared based on performance in word tasks in the sense of $g$ in (2). But there are several reasons why comparing performance in this way is difficult. First: performance may be affected less by the structure of model $f$, and more by the number of free parameters it entails and how well they have been tuned [Levy et al., 2015]. Second: for many embeddings, solving (1) entails a Monte Carlo optimisation, so different runs with identical $f$ will result in different realisations of $V$ and hence different values of $g(D, V)$. Third, more subtle and often conflated with the first and second: for most embedding models $f$, (1) does not uniquely identify $V$ — $V$ is said to be *non-identifiable* — and different solutions, $V$, each equally optimal with respect to (1), correspond to different values of $g(D, V)$.

This raises the disconcerting question: can apparent differences in performances in word tasks as evaluated with $g$ be substantially attributed to the arbitrary selection of a solution $V$ from the set of solutions of $f$? In this paper we explore the non-identifiability of $V$, particularly with respect to the class of non-singular transformations $C$ for which $f(X, UV) = f(X, UC^{-1}CV)$ but $g(D, V) \neq g(D, CV)$, and the consequences for constructing and evaluating word embeddings. Specifically, our contributions are as follows.

1. For $g$ defined using inner products of embedded word vectors (e.g. Cosine similarity) in $d$ dimensions, we characterise the subset $\mathcal{F}_d$ contained in the set of non-singular transformations to which $g$ is not invariant.

2. We study a widely used strategy for constructing word embeddings that involves multiplying a "base" embedding by a powered matrix of singular values, and show that this amounts to exploring a one-dimensional subset of the optimal solutions.

3. We discuss resolutions to the non-identifiability, including (i) constraining the set of solutions of $f$ to ensure compatibility with invariances of $g$, and (ii) optimizing over the solutions of $f$ with respect to $g$ in a supervised learning sense.

## 2 Non-identifiability of word embedding $V$

The issue of non-identifiability is most transparent in word embedding models explicitly involving matrix factorisation. LSA assumes $X$ is an $n \times p$ context-word matrix and seeks $V$ as

$$\underset{U,V}{\arg\min} \ f(X, UV) := \underset{U,V}{\arg\min} \ \|X - UV\|, \tag{3}$$

where $\| \cdot \|$ is the Frobenius norm, and $U$ is an $n \times d$ matrix of contexts to be estimated. For any particular solution $\{U^*, V^*\}$ of (3) $\{U^*C^{-1}, CV^*\}$ is also a solution, where $C$ is any $d \times d$ invertible matrix. The solution of (3) for $V$ is hence a set

$$\{CV^* : C \in \mathsf{GL}(d)\} \tag{4}$$

where $\mathsf{GL}(d)$ denotes the general linear group of $d \times d$ invertible matrices.

One way to find an element of the solution set (4) is by using the singular value decomposition (SVD) of $X$. The SVD decomposes $X$ as $X = A\Sigma B^T$ where $A$ and $B$ have orthogonal columns and $\Sigma$ is a diagonal matrix with the singular values in decreasing order on the diagonal. Then a rank $d$ matrix that minimizes $\|X - X_d\|$ is $X_d = A_d \Sigma_d B_d^T$ where $A_d$ and $B_d$ are matrices containing the first $d$ columns of $A$ and $B$ respectively, and $\Sigma_d$ is the $d \times d$ upper left part of $\Sigma$ [Eckart and Young, 1936]. Hence a solution to (3) is obtained by taking

$$U^* = A_d, \quad V^* = \Sigma_d B_d^T, \tag{5}$$

called by Bullinaria and Levy [2012] the "simple SVD" solution. Bullinaria and Levy [2012] and Turney [2013] have investigated the word embedding $V^* = \Sigma_d^{1-\alpha} B_d^T$ which generalises $V^*$ in (5) by introducing a tunable parameter $\alpha \in \mathbb{R}$, motivated by empirical evidence that $\alpha \neq 0$ often leads to better performance on word tasks. Such an embedding is perfectly justified, however, as an alternative solution

$$U^* = A_d \Sigma_d^{\alpha}, \quad V^* = \Sigma_d^{1-\alpha} B_d^T,$$

to (3), for any $\alpha \in \mathbb{R}$. We can hence interpret the tuning parameter $\alpha$ as indexing different elements of the solution set (4), each optimal with respect to the embedding model $f$, with $\alpha$ free to be chosen so that the word-task performance $g$ is maximized.

Indeed, by choosing the particular solution $V^*$ in (5), and setting $C = \Sigma_d^{-\alpha}$, we see that tuning $\alpha$ amounts to optimising over the one-parameter subgroup $\gamma(\alpha) := \Sigma_d^{-\alpha} \in \mathsf{GL}(d)$, a one-dimensional subset of the $d^2$-dimensional group $\mathsf{GL}(d)$ to which $V$ is non-identifiable. The motivation for restricting the optimisation to this particular subset is unclear, however. In fact, it is not clear that choice of the matrix of singular values $\Sigma_d$ in the subgroup $\gamma$ necessarily leads to better performance with $g$; Figure 2 in Section 4.2, demonstrates superior performance for alternate (but arbitrary) diagonal matrices for certain values of $\alpha$.

Yin and Shen [2018] (see also references therein) recognise "unitary [equivalently orthogonal] invariance" of word embeddings, explaining that "two embeddings are essentially identical if one can be obtained from the other by performing a unitary [orthogonal] operation." Here "essentially identical" appears to mean with respect to the performance evaluation, our $g$ in this paper. We emphasise the distinction between this and the non-identifiability of $V$, which refers to the invariance of $f$ to a (typically larger) class of transformations. The distinction was similarly made by Mu et al. [2019] who suggested modifying the embedding model $f$ such that the class of invariant transformations of $f$ and $g$ match. We briefly discuss further their approach later.

**Remark 1.** The foregoing discussion focuses on the LSA embedding model, $f$ in (3), in which the optimal embedding $V$ arises clearly from a matrix factorisation $X \approx UV$ with respect to Frobenius norm, and the non-identifiability is transparent. But other embedding models, including word2vec and GloVe, are defined by different $f$ yet share the same property that $V$ is non-identifiable, i.e. that the solution is defined as the set (4). Levy et al. [2015] have shown that word2vec and GloVe both amount to solving implicit matrix factorisation problems each with respect to a particular corpus representation $X$ and metric. To see this, and the consequent non-identifiability, it is sufficient to observe, as with the objective of LSA, that the objective functions of word2vec and GloVe involve matrices $U$ and $V$ appearing only as the product $UV$.

## 3 Effect of non-identifiability of embeddings on $g$

The word embeddings are evaluated on tasks on the test data $D$ using the function $g$, which typically is based on cosine similarity between elements of $\mathbb{R}^d$. Our focus will hence be on functions $g$ that depend on $V$ only through the cosine similarity between its columns.

The set of invariances associated with such $g$ consists of the group $c\mathsf{O}(d) := \{cQ \in \mathsf{GL}(d) : c \in \mathbb{R}, Q \in \mathsf{O}(d)\}$, where $\mathsf{O}(d)$ is the subset of orthogonal matrices $\{Q \in \mathsf{GL}(d) : Q^T Q = QQ^T = I_d\}$. This set also contains the set of scale transformations $c\mathcal{I} := \{cI_d : c \in \mathbb{R} - \{0\}\}$. $\mathsf{O}(d)$ relates to transformations that leave $\langle v_1, v_2 \rangle$ invariant; the scale transformation preserves the angle between $v_1$ and $v_2$.

Figure 1 (left) illustrates the incompatibility between invariances of $f$ and $g$. For embedding dimension $d = 2$, $v_i$ and $v_j$ are 2D embeddings of words $i$ and $j$ obtained from solving $f$ with respect to coordinate vectors $\{e_1, e_2\}$. For $Q \in \mathsf{O}(d)$, with respect to orthogonally transformed coordinates $\{Qe_1, Qe_2\}$, $Qv_i$ and $Qv_j$ are also viable solutions of $f$. A $g$ that depends only on $\cos(v_i, v_j)$ has the same value for $\cos(Qv_i, Qv_j)$. On the other hand, equally valid solutions $Cv_i$ and $Cv_j$ of $f$ with respect to nonsingularly transformed coordinates $\{Ce_1, Ce_2\}$ for $C \in \mathsf{GL}(d)$ lead to a different value of $g$ since $\cos(v_i, v_i) \neq \cos(Cv_i, Cv_j)$ unless $C \in c\mathsf{O}(d)$.

Thus with respect to the evaluation function $g$, each solution from the set $\{CV^* : C \in c\mathsf{O}(d)\}$ is equally good (or bad). However, since $c\mathsf{O}(d) \subset \mathsf{GL}(d)$, there still exist embeddings $CV^*$ which solve $f$ with $g(\cdot, CV^*) \neq g(\cdot, V^*)$. Such $C$ are precisely those which characterise the incompatibility between invariances of $f$ and $g$. One such example is the set of $C$ given by the one-parameter subgroup $\mathbb{R} \ni \alpha \mapsto \Lambda^\alpha$, where $\Lambda$ is a $d$-dimensional diagonal matrix with positive elements. This generalises the subgroup $\gamma(\alpha)$ discussed in §2, which is the special case with $\Lambda = \Sigma_d$. Figure 1 (right) illustrates the solution set and 1D subsets $\{\Lambda^\alpha V^*\}$ for different $\Lambda$ and particular solutions $V^*$. The discussion above is summarised through the following Proposition.

**Proposition 1.** *Let $V^*$ be a solution of (1). Then $g$ is not invariant to non-singular transforms $V^* \mapsto \Lambda^\alpha V^*$ for any $\alpha \in \mathbb{R}$ unless $\Lambda \in c\mathcal{I}$ for some $c \in \mathbb{R}$.*

The key message from Proposition 1 is: for $\alpha_1, \alpha_2 \in \mathbb{R}$, *comparison of performances of embeddings $\Lambda^{\alpha_1} V^*$ and $\Lambda^{\alpha_2} V^*$ using $g$ depends on the (arbitrary) choice of the orthogonal coordinates of $\mathbb{R}^d$.* Note however that the choice of the orthogonal coordinates does not have any bearing on $f$, and

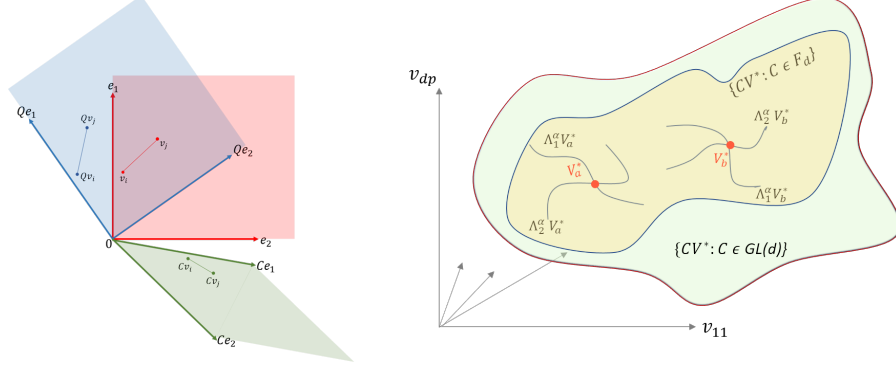

Figure 1: Left: For $d = 2$, orthogonally transformed coordinates $\{Qe_1, Qe_2\}$ (blue) with $Q \in \mathsf{O}(d)$, and nonsingularly transformed $\{Ce_1, Ce_2\}$ (green) with $C \in \mathsf{GL}(d)$, where $\{e_1, e_2\}$ (red) are standard coordinates. Distances between two embedding vectors $v_i$ and $v_j$ are preserved in the coordinates $\{Qe_1, Qe_2\}$, but altered in the coordinates $\{Ce_1, Ce_2\}$. However, $\{v_i, v_j\}$, $\{Qv_i, Qv_j\}$ and $\{Cv_i, Cv_j\}$ are valid solutions to (1). Right: Illustration of the solution set and one-dimensional subsets $\Lambda^\alpha V^*$ parameterised by $\alpha$ for two choices of $\Lambda$ and two particular solutions $V^*$.

hence $\Lambda^{\alpha_1} V^*$ and $\Lambda^{\alpha_2} V^*$ are both solutions of $f$. The first step towards addressing identifiability issues pertaining to $f$ and $g$ is to isolate and understand the structure of the set $\mathcal{F}_d$ of transformations in $\mathsf{GL}(d)$ which leave $f$ invariant but not $g$.

### 3.1 Structure of the set $\mathcal{F}_d$

What is the dimension of the set $\mathcal{F}_d \subset \mathsf{GL}(d)$? The dimension of $\mathsf{GL}(d)$ is $d^2$ and that of $\mathsf{O}(d)$ is $d(d-1)/2$. Since $c\mathcal{I}$ is one-dimensional, the dimension of $\mathcal{F}_d$ is $d^2 - d(d-1)/2 - 1 = d(d+1)/2 - 1$. Figure 1 (right) clarifies the implication of the result of Proposition 1: given a solution $V^*$, tuning $\alpha$ explores only a one-dimensional set within $\{CV^* : C \in \mathcal{F}_d\}$ (yellow) within the overall solution set $\{CV^* : C \in \mathsf{GL}(d)\}$ (green).

A group-theoretic formalism is useful in precisely identifying $\mathcal{F}_d$. Since $\mathsf{O}(d)$ is a subgroup of $\mathsf{GL}(d)$, we are interested in those elements of $\mathsf{GL}(d)$ that cannot be related by an orthogonal transformation. Such elements can be identified as the (right) coset $\mathsf{GL}(d) \setminus \mathsf{O}(d)$ of $\mathsf{O}(d)$ in $\mathsf{GL}(d)$: equivalence classes $[C] := \{QC : Q \in \mathsf{O}(d)\}$ for $C \in \mathsf{GL}(d)$, known as *orbits*, under the equivalence relation $M \sim N$ if there exists $Q \in \mathsf{O}(d)$ such that $M = QN$. The set of orbits $\{[C] : C \in \mathsf{GL}(d)\}$ forms a partition of $\mathsf{GL}(d)$: each nonsingular transformation $C \in \mathsf{GL}(d)$ is associated with its $[C]$, elements of which are orthogonally equivalent.

From the definition of $\mathsf{GL}(d) \setminus \mathsf{O}(d)$, we can represent $\mathcal{F}_d$ as $\mathcal{F}_d = \tilde{\mathcal{F}}_d - c\mathcal{I}$, where $\tilde{\mathcal{F}}_d$ represents what is left behind in $\mathsf{GL}(d)$ once $\mathsf{O}(d)$ has been 'removed', and $-$ denotes the set difference.

**Proposition 2.** *The set $\tilde{\mathcal{F}}_d$ can be identified with the subgroup $\mathsf{UT}(d)$ of upper triangular matrices within $\mathsf{GL}(d)$ with positive diagonal entries.*

*Proof.* The proof is based on identifying a set $S \subset \mathsf{GL}(d)$ that is in bijection with the orbits in $\mathsf{GL}(d) \setminus \mathsf{O}(d)$. Such a subset $S$ is known as a cross section of the coset $\mathsf{GL}(d) \setminus \mathsf{O}(d)$, and intersects each orbit $[C]$ at a single point. Since $\mathsf{O}(d)$ is a subgroup of $\mathsf{GL}(d)$, no two members of $\mathcal{F}_d$ belong to the same orbit $[C]$ of any $C \in \mathsf{GL}(d)$. Thus $\mathcal{F}_d$ can be identified with *any* cross section of $\mathsf{GL}(d) \setminus \mathsf{O}(d)$.

The map $\mathsf{GL}(d) \ni C \mapsto h(C) := C^T C$ is invariant to the action of $\mathsf{O}(d)$ since $h(QC) = (QC)^T QC = C^T C$. This implies that $h$ is constant within each orbit $[C]$. To show that $h$ is maximal invariant, we need to show that $h(C_1) = h(C_2)$ if and only if there is a $Q \in \mathsf{O}(d)$ with $C_1 = QC_2$. To see this, suppose that $C_1^T C_1 = C_2^T C_2$, and let $v_1, ..., v_d$ be a basis for $\mathbb{R}^d$. Let $x_i = C_1 v_i$ and $y_i = C_2 v_i$. Then $\langle x_i, x_j \rangle = \langle C_1 v_i, C_1 v_j \rangle = \langle v_i, C_1^T C_1 v_j \rangle = \langle v_i, C_2^T C_2 v_j \rangle = \langle C_2 v_i, C_2 v_j \rangle = \langle y_i, y_j \rangle$. There thus exists a linear isometry, say $Q$, such that $Qy_i = x_i$ for $i = 1, ..., d$. This implies that $QC_2 v_i = C_1 v_i$ for $i = 1, ..., d$, and since $v_1, ..., v_d$ is a basis for $\mathbb{R}^d$, $QC_2 = C_1$ with $Q \in \mathsf{O}(d)$. Thus the range of $h$ is in bijection with the orbits in $\mathsf{GL}(d) \setminus \mathsf{O}(d)$, and constitutes a cross section.

For any $C \in \mathsf{GL}(d)$ consider its unique QR decomposition $C = QR$, where $Q \in \mathsf{O}(d)$ and $R \in \mathsf{UT}(d)$, made possible since $R$ is assumed to have positive diagonal elements. Clearly then $h(C) = h(QR) = R^T R$, and its range $h(\mathsf{GL}(d))$ can be identified with the set $\mathsf{UT}(d)$. $\qquad\square$

**Remark 2.** The result of Proposition 2 can be distilled to the existence of a unique QR decomposition of $C \in \mathsf{GL}(d)$: $C = QR$, where $Q \in \mathsf{O}(d)$ and $R \in \mathsf{UT}(d)$. There is no loss of generality in assuming that $R$ has positive entries along the diagonal, since this amounts to multiplying by another orthogonal matrix which changes signs accordingly. Thus the map $\mathsf{GL}(d) \ni C \mapsto \{\mathsf{UT}(d) - c\mathcal{I}\}$ uniquely identifies an element of $\mathcal{F}_d$.

The map $\mathsf{GL}(d) \ni C \mapsto h(C) = C^T C$ is referred to as a maximal invariant function, and indexes the elements of $\mathsf{GL}(d) \setminus \mathsf{O}(d)$, and hence $\mathsf{UT}(d)$. This offers verification of the fact that the dimension of $\mathcal{F}_d$ is $d(d+1)/2 - 1$ since it is one fewer than the dimension of the subgroup $\mathsf{UT}(d)$. Another way to arrive at the conclusion is to notice that any $d \times d$ upper triangular matrix $R$ can be represented as $R = D(I_d + L)$, where $I_d$ is the identity, $L$ is an upper triangular matrix with zeroes along the diagonal, and $D$ is a diagonal matrix. The dimension of the set of $L$ is $d(d-1)/2$ and that of the set of $D$ is $d$, resulting in $d + d(d-1)/2 = d(d+1)/2$ as the dimension of the set of $R$.

## 4 Resolving the problem of non-identifiability

From the preceding discussion we gather that $\{CV^* : C \in \mathcal{F}_d\}$ comprises the set of solutions of $f$ which do not leave $g$ invariant. We explore two resolutions: (i) imposing additional constraints on $V$ in (1) to identify solutions up to $C \in \mathsf{O}(d)$ (Theorem 1), and uniquely (Corollary 1); and (ii) considering $C$ as a free parameter. In (i) the identified solution is chosen in a way that is mathematically natural, but need not be necessarily optimal with respect to $g$. In (ii), where $C$ is considered as a free parameter, it may be chosen to optimize performance in tasks, i.e., by optimising $g(D, CV^*)$ over $C \in \mathsf{UT}(d)$.

### 4.1 Constraining the solution set

Redefine (1) as a constrained optimisation

$$\underset{U,V:V\in\mathfrak{C}_v}{\arg\min}\ f(X, UV), \tag{6}$$

over a subset $\mathfrak{C}_v$ of possible values of $V$ which ensures that the only possible solutions are of the form $\{CV^* : C \in \mathsf{O}(d)\}$ for any solution $V^*$. The set of possible $U$ is unconstrained. From Proposition 2 and the QR decomposition of an element of $\mathsf{GL}(d)$, this is tantamount to ensuring that $CV^*$ for $C \in \mathsf{UT}(d)$ is a solution of (6) if and only if $C = I_d$, the identity matrix. Theorem below identifies the set $\mathfrak{C}_v$ for *any* solution of $U$.

**Theorem 1.** Let $\mathfrak{C}_v = \{V \in \mathbb{R}^{d \times p} : VV^T = I_d\}$. Then for any solution $V^*$ to the constrained problem (6), any other solution of the form $CV^*$ for $C \in \mathsf{GL}(d)$ satisfies $g(D, CV^*) = g(D, V^*)$ for a given test data $D$.

*Proof.* Let $\{\bar{U}, \bar{V}\}$ be a solution to the unconstrained problem. The proof rests on the simultaneous diagonalisation of $\bar{V}\bar{V}^T$ and $\bar{U}^T\bar{U}$. Since $\bar{V}\bar{V}^T$ is positive definite there exists $M \in \mathsf{GL}(d)$ such that $\bar{V}\bar{V}^T = M^T M$. Then $M^{-T}(\bar{U}^T\bar{U})M^{-1}$ is symmetric, and there exists $Q \in \mathsf{O}(d)$ such that $Q^T M^{-T}(\bar{U}^T\bar{U})M^{-1}Q = \Lambda$, where $\Lambda$ is diagonal. Setting $C = M^{-1}Q$ results in $C^T\bar{V}\bar{V}^T C = Q^T M^{-T}(\bar{V}\bar{V}^T)M^{-1}Q = I_d$.

We thus arrive at the conclusion that there exists a $C \in \mathsf{GL}(d)$ such that $C^T\bar{V}\bar{V}^T C = I_d$, and $C^T\bar{U}^T\bar{U}C = \Lambda$. The elements of $\Lambda$ solve the generalised eigenvalue problem $\det(\bar{U}^T\bar{U} - \lambda\bar{V}\bar{V}^T)$. Evidently then $C \in \mathsf{GL}(d)$ is orthogonal if $\bar{V}\bar{V}^T = I_d$. $\qquad\square$

An obvious but important corollary to the above Theorem is that any two solutions from $\mathfrak{C}_v$ are related through an orthogonal transformation (not necessarily unique).

**Corollary 1.** *For any solutions $V_1$ and $V_2$ of (6) in $\mathfrak{C}$ there exists an $Q \in \mathsf{O}(d)$ such that $QV_1 = V_2$. In other words, $\mathsf{O}(d)$ acts transitively on $\mathfrak{C}$.*

**Remark 3.** Optimisation over the constrained set $\mathfrak{C}_v$ results in a reduction of the invariance transformations of $f$ from $\mathsf{GL}(d)$ to $\mathsf{O}(d)$. This can be understood as choosing $CV^*$ for a fixed solution $V^*$ and arbitrary $C \in \mathsf{GL}(d)$, performing a Gram–Schmidt procedure to obtain $QRV^*$ for an $Q \in \mathsf{O}(d)$ and $R \in \mathsf{UT}(d)$, and discarding $R$. Topologically then, the set of solutions $\{QV^* : Q \in \mathsf{O}(d)\}$ is homotopically equivalent to the set $\{CV^* : C \in \mathsf{GL}(d)\}$. This is because the inclusion $\mathsf{O}(d) \hookrightarrow \mathsf{GL}(d)$ is a homotopy equivalence, as it is well-known that the Gram Schmidt process $\mathsf{GL}(d) \to \mathsf{O}(d)$ is a (strong) deformation retraction.

A unique solution for $V$ can be identified by imposing additional constraints on $U$ as follows.

**Corollary 2.** *Denote by $\mathfrak{C}_u$ the set of all $U \in \mathbb{R}^{n \times d}$ which satisfy the following conditions: (i) The columns of $U$ are orthogonal; (ii) the diagonal elements of $U^T U$ are arranged in descending order; (iii) first non-zero element of each column of $U$ is positive. Then, any solution to the optimisation problem in* (1) *over the constrained set* $(U, V) \in \mathfrak{C}_u \times \mathfrak{C}_v$ *is unique.*

*Proof.* We need to show that on the constrained space $\mathfrak{C}_u \times \mathfrak{C}_v$, the orthogonal $C$ obtained by optimising (6) reduces to the identity.

On the set $\mathfrak{C}_v$, from the proof of Theorem 1, we note that there exists a $C \in \mathsf{O}(d)$ such that $C^T \bar{U}^T \bar{U} C = \Lambda$ for a diagonal $\Lambda$ containing the eigenvalues of $U^T U$ with respect to $VV^T$ obtained a solution of $\det(\bar{U}^T \bar{U} - \lambda \bar{V} \bar{V}^T)$.

In addition to being orthogonal, condition (i) forces $C$ to be a matrix with each column and row containing one non-zero element assuming values $\pm 1$. In other words, $C$ is forced to be a monomial matrix with entries equal to $\pm 1$. This implies that the diagonal $C^T U^T U C$ contains the same elements as $U^T U$, but possibly in a different order. Condition (ii) then fixes a particular order, and condition (iii) ensures that each diagonal element is +1. We thus end up with $C = I_d$. $\qquad\square$

The idea to modify the optimisation so that the solution is unique up to transformations in $\mathsf{O}(d)$, but not necessarily $\mathsf{GL}(d)$, is also used by Mu et al. [2019]. Rather than place constraints on $V$, as above, they modified the objective $f$ to include Frobenius norm penalties on $U$ and $V$, which achieves the same outcome, although the relationship between the solutions of the penalised and unpenalised problems is not transparent.

### 4.1.1 Exploiting symmetry of $X$

If the corpus representation $X$ is a symmetric matrix, for example involving counts of word-word co-occurrences, then the rows of $U$ and the columns of $V$ both have the same interpretation as word embeddings. In such cases the symmetry motivates the imposition $U^T = V$. For example, in LSA (3) and its solution (5), this is achieved by taking $\alpha = 1/2$, since $A_d = B_d$ owing to the symmetry. This identifies a solution up to sign changes and permutations of the word vectors, transformations which are contained within $\mathsf{O}(d)$ and hence are of no consequence to $g$.

In GloVe, Pennington et al. [2014] observe that when $X$ is symmetric the $U^T$ and $V$ are equivalent but differ in practise "as a result of their random initializations". It seems likely that different runs involve the optimisation routine converging to different elements of the solution set, and not in general to solutions with $U^T = V$. For a given run Pennington et al seek to treat solutions $U^{*T}$ and $V^*$ symmetrically by taking the word embedding to be $V = U^{*T} + V^*$, which is not itself in general optimal with respect to the GloVe objective function, $f$ (although they report that using it over $V = V^*$ typically confers a small performance advantage). A different approach is to take the embedding to be $V = CV^*$ where $C \in \mathsf{GL}(d)$ is the solution to the equation $C^{-T} U^{*T} = CV^*$ which more directly identifies an element of the solution set for which $U^T = V$, and hence avoids taking the final embedding to be one that is non-optimal with respect to criterion $f$. The same strategy is also appropriate to other word embedding models, e.g. word2vec.

### 4.2 Optimizing over $\mathcal{F}_d$

To what extent can we optimize word-task performance $g(D, V)$ by choosing an appropriate element $V$ of the solution set (4)? The set of transformations $\mathcal{F}_d$ has dimension $d(d+1)/2 - 1$, typically much larger than the number of cases in $d$, so care is needed to avoid overfitting. In particular, if the embeddings generated are to be regarded as a predictive model, then it is necessary to use cross-validation rather than just optimising the embeddings with respect to a particular test set. One

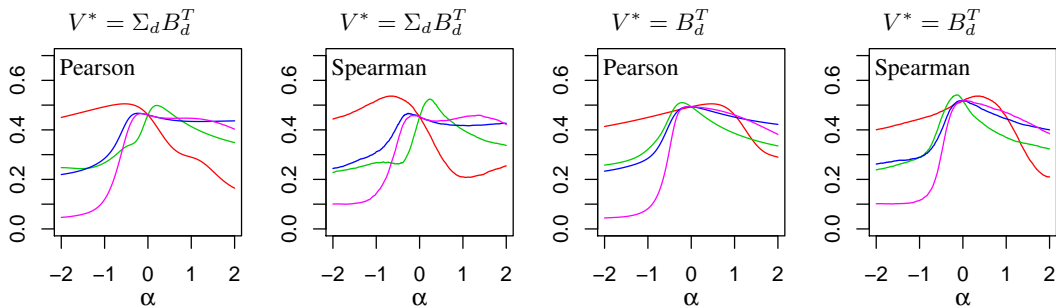

Figure 2: Plots showing word task evaluation scores $g(D, V)$ corresponding to the WordSim-353 task [Finkelstein et al., 2002] (located at `http://www.cs.technion.ac.il/~gabr/resources/data/wordsim353/`) which provides a set of word pairs with human-assigned similarity scores. The embeddings are evaluated by calculating the cosine similarities between the word pairs and using either Pearson or Spearman correlation (each invariant to $\mathsf{O}(d) \cup c\mathcal{I}$) to score correspondence between embedding and human-assigned similarity values. The embedding is from model (3), with $X$ taken to be a document–term matrix computed from the Corpus of Historical American English [Davies, 2012], and the plotted lines show how performance varies with different elements of the solution set, namely $V = \Lambda^\alpha V^*$ for $V^*$ as indicated and different $\Lambda = \text{diag}(\lambda_1, ..., \lambda_d)$ as follows: $\Lambda = \Sigma_d$ (red lines); $\lambda_i = i$ (green); $\lambda_i \sim U(0, 1)$ (blue); and $\lambda_i \sim |N(0, 1)|$ (purple). Performance for $\Lambda = \Sigma_d$, which is widely used, is not obviously superior to performance of the other completely arbitrary choices for $\Lambda$.

approach is to restrict the dimension of the optimisation, for example as earlier by considering solutions $V = \Lambda^\alpha V^*$ for a particular solution $V^*$ and diagonal matrix $\Lambda$. A widely used approach corresponds to choosing $\Lambda = \Sigma_d$, a matrix containing the dominant singular values of $X$; Figure 2 shows how $g$ varies with $\alpha$ for this $\Lambda$ and some other choices of $\Lambda$ chosen quite arbitrarily (details in the caption). There is clearly substantial variability in $g$ with $\alpha$, but performance with $\Lambda = \Sigma_d$ is only on a par with the other arbitrary choices.

Figure 3 shows the distribution of $g$ for $V = RV^*$ where $V^*$ is a GloVe embedding, and $R$ is a random element of $\mathcal{F}_d$, which is either upper triangular or diagonal, with its non-zero elements taken from the distribution $|N(0, 1)|$, and $g$ measures the performance of the embeddings on two similarity test sets. (More details in caption.) The histograms shows substantial variance in the scores for different $R$. The score for the base embedding $V^*$ is at the higher end of the distribution, though for some instances of random $R$ the performance of $V$ is superior. It is also noticeable that there is a much greater range of scores when $R$ is sampled from the set of diagonal matrices than when it is sampled from the set of upper triangular matrices. We hypothesize that this is because when $R$ is diagonal, there is a possibility of very small elements on the diagonal which will essentially wipe out whole rows of $V$, which could have a significant impact on the results.

Table 1 shows scores that result from optimising $g(D, V)$ for $V = \Lambda V^*$ with respect to the elements of $\Lambda = \text{diag}(\lambda_1, ..., \lambda_d)$, using R's `optim` implementation of the Nelder–Mead method, where $V^*$ are 300-dimensional embeddings generated using GloVe and word2vec. The results show that there exists a transformed embedding $\Lambda V^*$ that performs substantially better than the base embedding. In practice, in order to use this optimization method to generate embeddings, it would be necessary to use cross-validation, as embeddings which achieve optimal performance with respect to one test set may do less well on others. Our aim here is merely to point out that it is possible to improve the test scores by optimizing over elements of $\Lambda$.

## 5    Conclusions

We summarise our conclusions as follows.

1. Examining word embeddings — including LSA, word2vec, GloVe — through the relationship with low-rank matrix factorisations with respect to a criterion $f$ makes it clear that

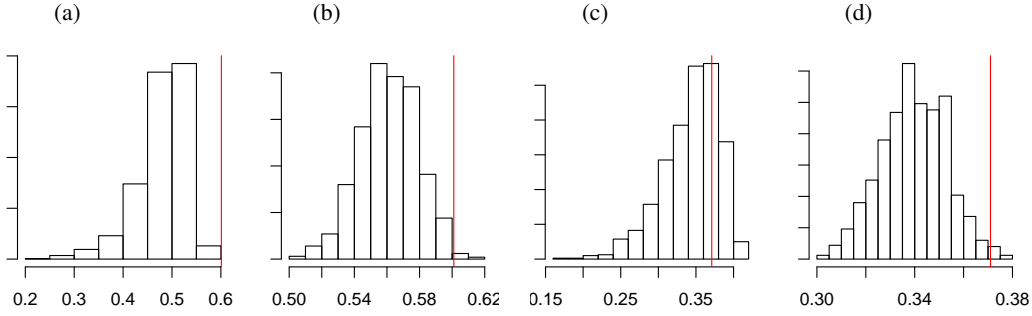

Figure 3: For the same type of task as in Fig. 2, histograms of Spearman correlation scores for embeddings $V = RV^*$ where $V^*$ is a GloVe embedding[1] with $d = 300$ trained on Wikipedia 2014 + Gigaword 5 corpus, evaluated on the WordSim-353 test set in (a) and (b), and on the SimLex-999 test set [Hill et al., 2015] in (c) and (d). $R \in \mathcal{F}_d$ is a random matrix, taken to be diagonal in (a) and (c) and upper-triangular in (b) and (d), in each case with the non-zero elements each distributed as $|N(0, 1)|$. The number of runs in each case was 1000. The red line on each graph shows the score for the original embedding in each case. [1]Source: https://nlp.stanford.edu/projects/glove/

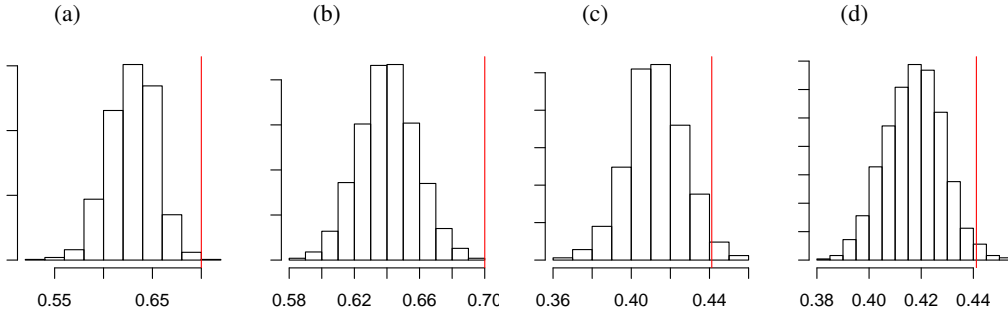

Figure 4: Histograms showing the performance of word2vec embeddings trained on the 100-billion word Google News corpus, where $d = 300$ (downloaded from https://code.google.com/archive/p/word2vec). As for Figure 3, the test set used is the WordSim-353 test set in (a) and (b), and SimLex-999 in (c) and (d), with the test score being calculated using the Spearman correlation coefficient. In graphs (a) and (c) $R$ is sampled from the set of diagonal matrices, and in (b) and (d) it is taken to be upper triangular.

the solution $V$ is non-identifiable: for a particular solution $V^*$, $CV^*$ for any $C \in \mathsf{GL}(d)$ is also a solution. Different elements of the $d^2$-dimensional solution set perform differently in evaluations, $g$, of word task performance.

2. An important implication is that the disparity in performance between word embeddings on tasks $g$ maybe due to the particular elements selected from the solution sets. In word embeddings for which the $f$ is optimized numerically with some randomness, for example in the initializations, the optimisation may converge to different elements of the solution set. An embedding chosen based on the best performance in $g$ over repeated runs of the optimisation can essentially be viewed as a Monte Carlo optimisation over the solution set.

3. The evaluation function $g$ is usually only invariant to orthogonal ($\mathsf{O}(d)$) and scale-type ($c\mathcal{I}$) transformations. Thus for an embedding dimension $d$, the effective dimension of the solution set after accounting for the orthogonal transformations, and scaled versions of the identity, is $d(d+1)/2 - 1$. Conclusions from evaluations with large $d$ must hence be interpreted with some care, especially if the $V$ is optimized with respect to the incompatible transformations $\mathcal{F}_d$ directly or indirectly, for example as in point 2 above.

4. These considerations have a bearing on the interpretation of the performance of the popular embedding approach of taking $V = \Lambda^\alpha V^*$ where $\alpha$ is a tuning parameter and $\Lambda$ is a diagonal

| Test set | Embeddings | Spearman | Pearson |
|---|---|---|---|
| WordSim-353 | GloVe vectors reported in [Pennington et al., 2014] | 0.658 | |
| | GloVe embedding $V^*_{\text{GloVe}}$ | 0.601 | 0.603 |
| | GloVe embedding $V^*_{\text{GloVe}}$ with Equation 6 imposed | 0.641 | 0.637 |
| | $V = \Lambda V^*_{\text{GloVe}}$ optimized over $\Lambda = \text{diag}(\lambda_1, ..., \lambda_d)$ | 0.679 | 0.760 |
| | word2vec embedding $V^*_{\text{w2v}}$ | 0.700 | 0.652 |
| | word2vec embedding $V^*_{\text{w2v}}$ with Equation 6 imposed | 0.645 | 0.588 |
| | $V = \Lambda V^*_{\text{w2v}}$ optimized over $\Lambda = \text{diag}(\lambda_1, ..., \lambda_d)$ | 0.797 | 0.838 |
| SimLex-999 | GloVe embedding $V^*_{\text{GloVe}}$ | 0.371 | 0.389 |
| | GloVe embedding $V^*_{\text{GloVe}}$ with Equation 6 imposed | 0.402 | 0.421 |
| | $V = \Lambda V^*_{\text{GloVe}}$ optimized over $\Lambda = \text{diag}(\lambda_1, ..., \lambda_d)$ | 0.560 | 0.582 |
| | word2vec embedding $V^*_{\text{w2v}}$ | 0.441 | 0.453 |
| | word2vec embedding $V^*_{\text{w2v}}$ with Equation 6 imposed | 0.475 | 0.480 |
| | $V = \Lambda V^*_{\text{w2v}}$ optimized over $\Lambda = \text{diag}(\lambda_1, ..., \lambda_d)$ | 0.583 | 0.617 |

Table 1: Evaluation task scores $g(D, V)$ corresponding to WordSim-353 [Finkelstein et al., 2002] and SimLex-999 [Hill et al., 2015] test sets. The base GloVe embedding $V^*$ is as described in the caption of Figure 3; the word2vec embedding is as described in the caption of Figure 4.

In the first row we note for reference the performance reported in [Pennington et al., 2014]. The results indicate substantial scope for improving performance scores via an appropriate choice of $\Lambda$.

matrix taken, for example, to contain the singular values of $X$. This amounts to providing a way to perform a search over a one-dimensional subset of the $(d(d+1)/2-1)$-dimensional solution set. Our numerical results suggest there is nothing special about this particular choice of $\Lambda$ (or the corresponding one-dimensional subset being searched over), nor is there a clear rationale for restricting to a one-dimensional subset.

## Acknowledgments

The authors gratefully acknowledge support for this work from grants NSF DMS 1613054 and NIH RO1 CA214955 (KB), a Bloomberg Data Science Research Grant (KB & SP), and an EPSRC PhD studentship (RC).

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
