[Reviews · NeurIPS 2019]

Reviewer 1



This paper was very clear, easy to follow and tackled an important topic under a new perspective. It gives a lot of insight on how embeddings are trained and evaluated, opening up space and motivating new research on this topic. Nonetheless, the paper needs more details on the experiments, I couldn’t understand on which data embeddings were optimized, for example. The paper should also give a clearer motivation for the choice of how embeddings were constrained in Section 4.1. In the moment I do not know if constrained embeddings would provide better, worse or similar results to the average non constrained ones. As an extra, I believe a few extra experiments (showing results for other embeddings, for example) would help. They would give a more palpable notion of how large the impact of varying embeddings could be. *What strengths does this paper have?* The paper is very well written. Although it is very mathematically grounded, it is easy to follow and understand. I believe even readers with a reduced mathematical understanding could (although maybe skipping the proofs) understand the paper. It explores a non-mainstream topic in a relevant task. It gives interesting insights on this topic which other work can build on. It proposes two different solutions to the problems they highlight in the paper. *What weaknesses does this paper have?* The paper states that there is a discrepancy between invariant regions of f and g. One of their solutions is to restrict the set of solutions V is allowed to take. Nonetheless, the authors do not show why this reduced set of solutions for V would be better than a random choice in the full set of optimal results for f. They also do not give evaluation scores of g to this restricted V embedding choice. This could be presented in Table 1. A second problem is that the authors do not give a detailed description of how \Lambda is optimized for the results in Table 1. Do they optimize this matrix on the same data D used later to get the scores g(D, V). If so, I believe this would be a critical problem. This optimization should be made on a set of data D’ which does not intercept with D. The paper had some extra space left and the code does not seem hard to test with other embeddings. Why not expand Table 1 to contain results for LSA and Word2Vec? *Detailed comments:* In Figure 1 (a, b), the red lines do not match the others when “alpha=0”. I imagine this is because you didn’t do “\Lambda ^ alpha * \Sigma * B” for this line, but only “\Lambda ^ alpha * B”. If this is true, I believe these plots would be more intuitive if it used the full option (“\Lambda ^ alpha * \Sigma * B”). In Figure 3, results for upper triangular matrices (b, d) seem to usually have better performance than diagonal ones (a, c). Would you have intuitions on why this happens? What is the red vertical line in these plots? Original V results? Typos or small comments: Line 116: Was the difference supposed to be between != ? Or was it supposed to be != ? Line 219: In addition to *being* orthogonal… Figure 2 and 3: These figures are leaking outside the margin. Figure 2: A legend could be presented visually together with the plots in someway. Figure 3: There is no xlabel and there is no range for the y axis.

Reviewer 2



The authors studies the inconsistency between the training and evaluation of word embeddings. Specifically, the training phase consists of an optimization procedure that usually involves a low-dimension approximation of a representation matrix (e.g. the word co-occurrence matrix). In the evaluation phase, the embeddings are evaluated against an objective that is usually unitary-invariant (e.g. involving only inner-product of vectors). The inconsistency between the training and evaluation objectives has to aspects: - Without any additional constrain or regularization of the training objective, the obtained embeddings are usually non-unique (the identifiability problem). - More importantly, since the training and evaluation phases are completely detached from each other, it is actually not clear why word embeddings should work for these evaluation tasks (the "meta" problem to which I am always craving for an answer!). The authors primarily studied the identifiability problem, and noticed that the non-identifiability could potentially cause problems. I appreciate that a mathematical characterization is presented. Yes, if we look at the training objective alone, it has more "degrees of freedom" than the evaluation objective, which is only subject to invariance under unitary and constant multiplications. This is a discrepancy. Whenever there is a discrepancy the natural question to ask is: which side should we fix? Evaluation is what it is, and it's our actual goal (we want embeddings to work well on them). In my opinion I do believe the non-identifiability issue is an artifact of the training objective function, which could be quickly fixed with some additional constrains or regularizations. Having an objective like ||X-UV|| is certainly not enough as V can be any full-rank matrix spanning the same column subspace, and this is clearly undesirable (for example, one can pick weird ones like the first column has very small norm and last column has very large norm). The previous methodologies (like skip-gram, LSA or word2vec) implicitly addresses this issue already, as they place implicit constrains (like requiring U and V to be -- either exactly or statistically -- symmetric). In the paper the authors made this point more explicit. The paper made a niche contribution in analyzing this identifiability issue. But I feel overall the part that is lacking is the "so what?" question. As previous methodologies already implicitly constrained themselves to avoid this identifiability issue (or more precisely, the incompatibility issue as their embeddings are still non-unique up to only unitary-transformations, which the evaluation objective does as well), it is hard to come up with an established example that are severely screwed by this issue. So in general I feel it can be more productive in looking for examples where - we can benefit by looking at a carefully selected larger space (a good example is "Uncovering Divergent Linguistic Information in Word Embeddings with Lessons for Intrinsic and Extrinsic Evaluation" which won the CoNLL '18 best paper, although they are more empirically focused), or - how to modify the training objective function to make the solutions compatible with the evaluation objective (i.e. we want solutions to be identifiable up to only the transformations we want, namely unitary transformations and constant multiplications), and - whether this can lead to new embedding methodologies that are theoretically more sound and performs better on these evaluation objectives? There are a few examples in the paper discussing the above points (in Section 4) and summarized a few strategies (constraining and exploit symmetry, which people are using already), but I feel there can be more, especially in terms of new methodologies and experiments. This was the point I enjoyed about the Levy and Goldberg "Neural Word Embedding as Implicit Matrix Factorization" paper; they did not stop at the analysis part, but they actually proposed new methods based on their analysis and showed that they outperform the old ones on a few tasks. I appreciate the fact that the authors made the non-identifiability issue explicit. On top of this, it will be great if we can see what we learned and how the methods should evolve. Towards this end, I like Section 4.2 since it provides a new idea, but I feel it is still some distance away from being full-fledged. The authors did not propose a systematic approach and the experiments seems a bit ad-hoc. ############################################################ I've read the rebuttal. The authors addressed my concerns reasonably well. By "symmetric" I meant that they are statistically equivalent, meaning the objective will not change if we re-denote U as V and V as U, the entire procedure is statistically identical (hence they should have same singular values, for example). In the matrix factorization scenario, this effectively requires that if M=UV^T, then U and V must each share half of the spectrum. Or, think about the following scenario: the objective in word2vec only concerns u^Tv; as a result, a reasonable learning algorithm (like SGD) will treat u and v equally, acting effectively as an implicit regularization.

Reviewer 3



This paper addresses an interesting problem in word embedding: given a downstream *WORD-task* and its evaluation metric g, which subset of solutions of the original word embedding (WE) learning problem is performance-invariant to g? And can we improve WE's performance with respect to g? Originality and Quality: The paper's analysis is relatively novel and insightful. To answer the raised questions, the paper considers a rather special linear case: Latent Semantic Analysis (LSA) with the specific inner product metric g = <*, *>. The author provides a mathematical analysis about which transformation groups {C} to a WE solution V* will stay performance-invariant with respect to g and the reasoning given is solid. The author also conducts an investigation about the non-invariant transformation groups and discuss some of their mathematical properties (although not all of them are necessary). In a word, Section 2 and 3 spark some deeper understanding of how a transformation on an existing WE solution will influence (or not influence) the g in WORD-tasks. The writing is good and easy to follow. Weakness: (1) Section 4.2 is a bit short and weak. Before reading this section I was expecting something like a more general solution rather than some short discussion about the existing special and simple method. (2) It would be great if the author could give further analysis (or at least a )that how the variance of different word embedding solutions may influence the performance in more complex *NON-WORD* tasks (not just word similarity measurement using <.,.>) as they are much more popular scenarios in practice.

[Author Response · NeurIPS 2019]

We are very grateful to the three reviewers for their careful reading of our paper, and for their insightful comments and
suggestions. Our responses are below with the reviewers' comments in italics.

• **Reviewer 1** speaks very positively about our paper tackling an *"important topic under a new perspective"*, and we
too anticipate that it will *"open up space and motivate new research"* in this area. In response to specific points:

– *"The paper needs more details on the experiments..."* Although we do give full details (of embedding models,
training data, embedding dimension, etc) in the Figure captions we appreciate that these might have gotten
somewhat lost; we will revise to clarify, and expand explanation in the main text.

– *"Give a clearer motivation for how embeddings were constrained in §4.1...the authors do not show why this
reduced set of solutions would be better than a random choice in the full set [nor give results for it, which] could
be presented in Table 1...Why not expand Table 1 to [include LSA, Word2Vec]".* As pointed out in Remark 3, the
constraints in §4.1 are related to the widely used Gram-Schmidt process, and are natural under the group-theoretic
formalism. We make no claim that this particular solution subset is optimal (or even at all good). We will amend
the text to make this clear, and also follow the excellent suggestion to add results corresponding to §4.1, as well as
Word2Vec, to Table 1.

– *"Do they optimize [$\Lambda$] on the same data $D$ used later to get the scores?"* We did not use cross-validation for the
results in Table 1, because the purpose of these results is to understand how much $g$ can be increased without
changing $f$, as we say in the caption to indicate the "substantial scope for improving performance scores via an
appropriate choice of $\Lambda$." We will amend presentation to further emphasise this, and also expand our caution of
overfitting (line 250) to discuss specifically the need for cross-validation when producing predictive models.

– *"Fig 2 (a,b), the red lines do not match when $\alpha = 0$...".* Thank you for pointing this out (and conjecturing correctly
the reason!). We will amend.

– *"Fig 3, results for upper triangular...have better performance than diagonal ones...[Intuition?]...What are red
lines...original V results?"* Our intuition here is that the highest performing instances do (slightly) better for the
upper triangular case than the diagonal case since the extra degrees of freedom give greater scope for a solution
with large $g$; and that the left tail is long because with |N(0,1)| random diagonal elements there is a substantial
chance of small elements that essentially wipe out whole rows of $V$. We will add this discussion in the revision.
The red lines are indeed the results for the original $V^*$, which we will now state in the caption.

– *Various typos:* Thank you for pointing these out, all of which we will fix.

• We share **Reviewer 2**'s craving in answering the "meta question" of *why* embeddings work. Given the impressive
success of numerous embedding methods being used and compared, we investigate a related, albeit modest, meta
question 'for a fixed evaluation function, can we identify properties/issues of objective functions that drive the
apparent disparity in performances of embedding methods?'. To our knowledge we are the first to do this.

– *"Previous methodologies...implicity address [non-identifiability] already, as they place implicit constraints (like
requiring U and V to be...symmetric)...[hence] 'so what?'"* It is not clear what implicit constraints the reviewer is
referring to in general. But U and V are non-square so cannot be symmetric. Perhaps "requiring U and V to be
identical on account of X being symmetric" was intended, as in §4.1.1. But this strategy is limited to situations
with X symmetric. As we discuss in §4.1.1, the remedy suggested by the authors of GloVe is ad hoc (much less
natural than the alternative we suggest) and leads to embeddings that are not even optimal with respect to the
model's own objective!
Our view is that to even begin to address the reviewer's "meta question" the first step is to understand embedding
methods' objective functions and the properties of embeddings that result. We feel it is paramount to understand
the impact of the identifiability issue first, instead of adding to the growing list of embedding methods by throwing
out new objective functions.

– Suggestions: (i) *"looking at a carefully selected larger space, [e.g. Artetxe et al 2018]".* The linear transform in
their paper (in our notation $V = \Lambda^\alpha Q V^*$) is just identified with a one-dimensional subset of $GL(d)$, and since $g$ is
invariant to any $Q \in O(d)$ their approach is covered by our Proposition 1. (ii) *"modifying the training objective".*
Addressed previously. (iii) *"embedding methodologies that are theoretically more sound and performs better..."* It
is unclear what the reviewer means by 'theoretically more sound'. Perhaps it's related to the suggestion in (ii)?

• **Reviewer 3** makes some helpful suggestions about expanding §4.2, in particular to include *"NON-WORD-tasks (not
just word similarity...using [inner product])".* We emphasise it is not just similarity but analogy tasks too for which $g$
involves $V$ only via the inner product of its columns, though we acknowledge the lack of analogy results, which some
readers may find more interesting, in §4 and will include some in the revised version. One point of clarification: the
reviewer notes that we focus on the *"rather special linear case [of] LSA"*, though we do so only in §2 as a particular
example that aids exposition of the identifiability issue; the paper's key results in §3 and §4 are appropriate to any
model of the very general form (1), including GloVe and word2vec.

[Meta-Review · NeurIPS 2019]

This paper is trying to provide some additional understanding of word embeddings. Given the ubiquity of word embeddings, we need more work like this. Even though the reviewers are not very confident, I support acceptance of this paper.